# Chiroptical Symmetry Analysis of Trianglimines: A Case Study

**Ani Ozcelik [1]**, **Raquel Pereira-Cameselle [1]**, **Ricardo A. Mosquera [2]**,
**Ángeles Peña-Gallego [2],\*** and **J. Lorenzo Alonso-Gómez [1],\***

[1]    Departamento de Química Orgánica, Universidade de Vigo, Lagoas-Marcosende s/n, 36310 Vigo, Spain;
       ozcelik@uvigo.es (A.O.); raquel@uvigo.es (R.P.-C.)
[2]    Departamento de Química Física, Universidade de Vigo, Lagoas-Marcosende s/n, 36310 Vigo, Spain;
       mosquera@uvigo.es
\*     Correspondence: mpena@uvigo.es (Á.P.-G.); lorenzo@uvigo.es (J.L.A.-G.)

**Abstract:** It is well established that chiroptical responses, based on the unique reaction to circularly polarized light by chiral non-racemic systems, are sensitive to the stereochemistry of the featuring systems. This behavior has promoted the use of chiroptical spectroscopies as a mandatory tool in the structure determination of molecules for decades. Recently, the higher sensitivity of chiroptical techniques compared to the conventional UV/Vis absorption and fluorescence spectroscopies or electrochemistry has awakened much interest in the development of chiroptical everyday applications. While chiroptical responses could be predicted by ab initio calculations, large systems calculated at a high level of theory may have an important computational cost; therefore, more intuitive methods are desired to design systems with tailored chiroptical responses. In this regard, the exciton chirality method has been often used in conformationally stable systems incorporating at least two independent chromophores. Taking this method into consideration, in our previous work, we described the chiroptical symmetry analysis (CSA) based on symmetry selection rules. To explore the scope of the CSA, herein we perform the chiroptical symmetry analysis of diverse trianglimines and draw general conclusions to assist on the design of chiroptical systems with high symmetry.

**Keywords:** chiroptical responses; high symmetry; exciton coupling; theoretical prediction

## 1. Introduction

An object is chiral when it is non-superimposable with its mirror-image [1]. Examples presenting this property vary from macroscopic such as screws or spiral staircases to microscopic i.e., nanoparticles [2] and molecules [3] among others. Furthermore, two chiral mirror-image molecules are called enantiomers. While the interaction between two enantiomeric systems and any achiral medium is indistinguishable, the behavior may differ when the two enantiomers interact with a chiral medium, giving rise to chiroptical responses when interacting with chiral light [4]. Electronic and vibronic circular dichroism (ECD and VCD) or optical rotatory dispersion (ORD) are some of the most common spectroscopies arising from such responses [5]. In the last few decades, these and other chiroptical spectroscopies have been widely employed for the structure determination of chemical systems [6]. On the other hand, the current interest in chiroptical light-powered molecular motors [7] or sensors [8] has triggered the development of efficient chiroptical systems [9–11]. Accordingly, the prediction and understanding of the chiroptical responses is essential [12].

Ab initio simulations have been shown to have great relevance for the prediction of chiroptical responses [13]. However, even though simplified time-dependent density functional theory (TD-DFT) techniques have been developed for large systems [14,15], the necessary computational and time

resources may be a limitation. As a qualitative alternative, the application of the exciton chirality method (ECM) [16] has been considered to be practical for systems incorporating two identical chromophores—for instance, molecular tweezers used as chiroptical proof of natural amines [17]. When two identical chromophores are in close vicinity, in phase and out of phase excitations arise. If the mutual orientation between the chromophores is chiral, the rotatory strength (RS) of the two transitions may be of opposite sign. The sign of the RS associated with the less energetic transition determines the sense of chirality [18]. Nevertheless, the presence of more chromophores may be desired for designing powerful chiroptical systems [19]. In this respect, we previously revised the chiroptical symmetry analysis (CSA) for systems with three and four chromophores [20]. The main scope of this approach is to serve as a useful tool for developing particular systems rather than predicting experimental spectra of systems. Like EC, CSA also provides the sense of chirality of the responding system. On the other hand, different parameters (i.e., rotatory strength or energy difference between allowed transitions) can be calculated by the CSA. Considering the conformational stability typically presented by trianglimines as well as taking advantage of the experimental and simulated ECD reported by Szymkowiak et al. [21], in this work, we perform the chiroptical symmetry analysis for 1, 2, and 3 (Figure 1) as a case study and intend to draw some conclusions about the reliability of this strategy on the design of tailored chiroptical systems.

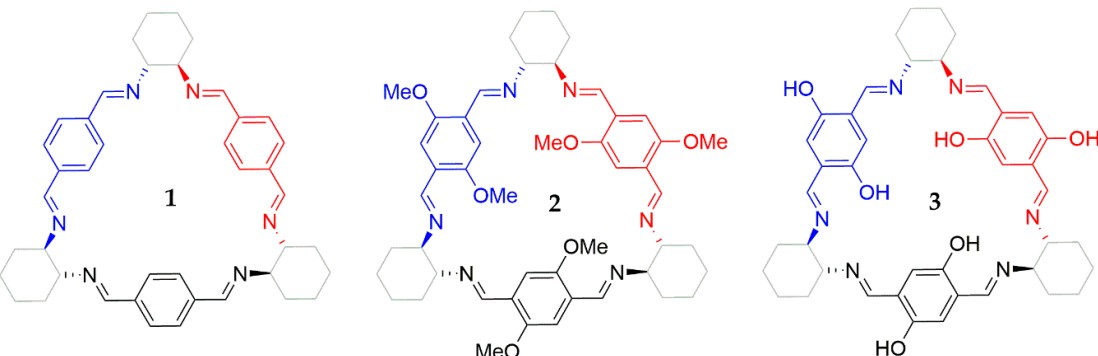

**Figure 1.** Trianglimine derivatives 1, 2, and 3 under study. The three equivalent chromophores in each structure are depicted in blue, red and black.

## 2. Materials and Methods

All calculations were performed with Gaussian09 program [22]. Trianglimine structures **1**, **2**, and **3** (Figure 1) were optimized at cam-B3LYP [23,24] level using 6-311G(d,p) basis set. Subsequently, vibrational frequencies were calculated at the same level of theory to confirm that the obtained structures were true minima. The structural measures necessary for the CSA were extracted from the corresponding structures (Figures 2 and 3). In addition, TD-DFT calculations were performed to compare the ECD spectra with those predicted by CSA. Particularly, the cam-B3LYP functional was chosen since the simulated spectra showed good agreement with the reported experimental ECD spectra [21]. On the other hand, UV/Vis and ECD spectra were simulated for analogue structures featuring only two and one chromophores (depicted in Figures 2 and 3, respectively) by removing the rest of the structure and adding hydrogen atoms for each analogue without further optimization.

All trianglimines under study have $D_3$ symmetry; therefore, the expressions obtained previously [20] for this symmetry were employed to predict the ECD spectra in a qualitative manner. The first step in the application of the CSA involves the identification of the chromophores that are responsible for optical properties in a molecule. Concerning thee trianglimine derivatives under study, three identical chromophores are present in each structure (represented in different colors in Figure 1). According to the CSA, the chiroptical response of a system mainly originates from the contribution of each chromophore interacting cooperatively with the others due to their relative orientation. This implies that the symmetry of the system determines which chromophores interact simultaneously

resulting in the ECD spectra. When a system with $C_2$ or $D_2$ symmetry presents two non-conjugated chromophores, the Davydov model allows for obtaining a picture of the approximate ECD spectra: The chromophores are represented by the corresponding Electric Dipole Transition Moment (EDTM), $\overrightarrow{\mu}_i^t$, and the interaction between them, $V_{12}$, is given by Equation (1):

$$V_{12} = R_{12}{}^{-3}\left[\overrightarrow{\mu}_1^t \cdot \overrightarrow{\mu}_2^t - 3R_{12}{}^{-2}\left(\overrightarrow{\mu}_1^t \cdot \overrightarrow{R}_{12}\right)\left(\overrightarrow{\mu}_2^t \cdot \overrightarrow{R}_{12}\right)\right]. \tag{1}$$

In this equation, $\overrightarrow{R}_{12}$ is the vector connecting the origin of $\overrightarrow{\mu}_1^t$ and $\overrightarrow{\mu}_2^t$ (Figure 2) and $V_{12}$ is in au units. The application of the perturbation theory to the originally degenerate first electronically-excited level yields an energy splitting, $\Delta E$, between these two states and, consequently, the gap between the two main electronic excitations given by Equation (2):

$$\Delta E = 2V_{12}. \tag{2}$$

The CSA seeks to obtain the splitting between those first electronically-excited states attainable from the ground state by absorption of electromagnetic radiation allowed by electric dipole transition. Additionally, the corresponding RS for systems with two or more independent chromophores could be revealed by this method. It should be noted that the application of CSA for systems bearing two chromophores is equal to that of ECM.

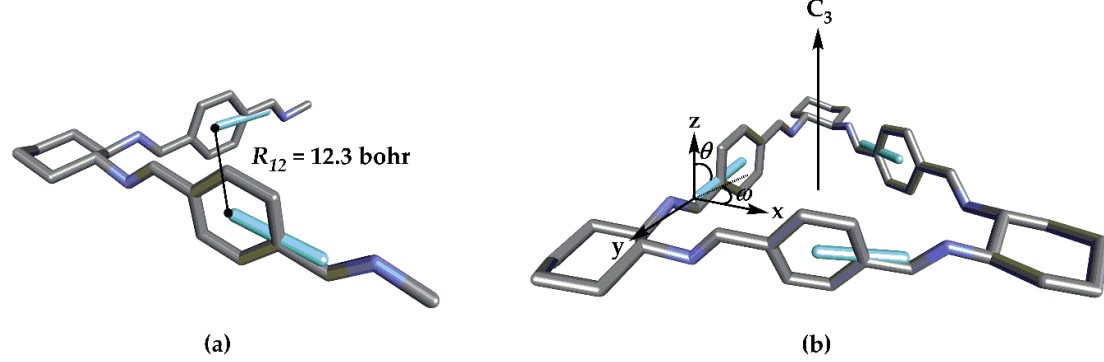

**Figure 2.** Structural measures of **1**. (**a**) $\overrightarrow{R}_{12}$ vector is extracted from analogue structure of **1** with two chromophores (**b**) while $\theta$ is the angle between Electric Dipole Transition Moment (EDTM) of chromophores and $C_3$ axis, $\omega$ stems from the angle between the projection of EDTM for chromophores on the $x$–$y$ plane with the $x$-axis. Light blue bars represent the EDTM associated with each independent chromophore.

The structural parameters can be obtained after determining the location of the EDTM in each chromophore. This will depend on the nature of the chromophore and may present more than one EDTM at different energies and orientations. Figure 3 presents the different EDTMs for the chromophores in 1, 2, and 3. While there is only one significant electronic transition for the chromophoric units in 1 (a), there are two in the case of 2 (2.1, b) and (2.2, c) and three for 3 (3.1, d), (3.2, e), and (3.3, f).

For systems presenting $D_3$ symmetry, the ground electronic state belongs to the totally-symmetric representation (A1) and there are three equivalent monoexcitations. With this basis set, we are able to obtain the $D_3$ Symmetry-Adapted Linear Combinations for the first electronically-excited states. As we have shown earlier [20], the reducible representation of this basis can decompose into A1 and E when all the molecular orbitals (MOs) describing the excited state are σ or A2 and E when the excited state of the chromophore contains one π MO. In any case, the transition from the ground state will be only orbitally allowed for states A1 and E or A2 and E with $3V_{12}$ energy difference between A1 or A2 and E.

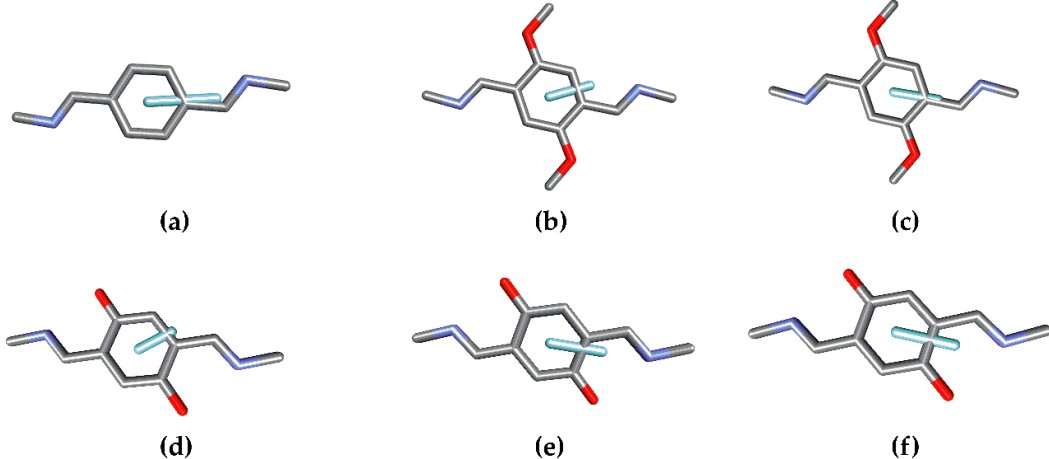

**Figure 3.** Representation of EDTMs for analogue structures with only one chromophore in 1 (**a**); 2.1 (**b**); 2.2 (**c**); 3.1 (**d**); 3.2 (**e**); and 3.3 (**f**) simulated by TD-DFT calculations.

The EDTM corresponding to the A2 transition is along the main symmetry axis, whereas the EDTMs for E transitions are located on the orthogonal plane. The rotatory strengths are obtained from Equations (3) and (4). It is noteworthy that this approximation is only valid when the electronic transition of the independent chromophore presents negligible magnetic transition dipole moment (MTDM, for a more thorough explanation see reference [20]). Other studies have developed ab initio based exciton chirality methods [25]:

$$R^{A2} = -3\pi \bar{v}^t R \left| \vec{\mu}_1^t \right|^2 \cos(\theta) \sin(\theta), \tag{3}$$

$$R^{E1} = R^{E2} = -\frac{1}{2} R^{A2}. \tag{4}$$

After this brief description of the CSA for molecules with $D_3$ symmetry, we need to know the structural parameters of the molecule as well as the EDTM of the monomer in order to use CSA on each particular case. By this way, after calculation of $V_{12}$ and the rotatory strengths for each allowed transition, we will be able to predict the main features of the ECD spectra originated from the exciton coupling between the interacting independent chromophores. Therefore, we will predict one A2 and two degenerated E transitions per electronic transition of the independent chromophores. Since the chromophoric units in 1, 2, and 3 present respectively one, two, and three significant electronic transitions, by the CSA, we could simulate one, two, and three A2 along with one, two, and three pairs of E transitions, respectively.

## 3. Results

Structures **1**, **2**, and **3** presenting three chromophores were hereby studied by CSA. Szymkowiak et al. demonstrated that TD-DFT calculations resemble the experimental ECD spectra of those compounds well [21]. Therefore, as mentioned above, DFT simulations were performed and compared with those obtained by CSA. As the EDTMs are origin independent, we can define an EDTM vector for each chromophoric transition (Figure 3). The molecular EDTM was obtained as a combination of individual vectors for each particular electronic transition to visualize the CSA and compared to those obtained from TD-DFT.

### 3.1. Structure 1

The different structural measures extracted for CSA analysis of structure **1** are presented in Table 1, whereas Figure 2 shows structure **1** and a truncated system with only two chromophores. Regarding

structure **1**, the analysis was also performed for the analogue with only two chromophores as an example of application of the CSA. In the case of two-chromophore systems, two excited states result from the parallel and antiparallel combinations.

**Table 1.** Structural measures for CSA analysis of **1**. As illustrated in Figure 2, $R_{12}$ is the distance between origins of the ETDMs of two chromophores, $\theta$ is the angle between EDTM and $C_3$-axis ($z$-axis) and $\omega$ is the angle between the projection of EDTM for chromophores on the $xy$ plane with the $x$-axis (for $D_3$ symmetry, the required angle is 90°). $V_{12}$ value is obtained as indicated [3].

| Structure | $R_{12}$ | $\omega$ | $\theta$ | $V_{12}$ |
|---|---|---|---|---|
| 1 [1] | 12.3 bohr | 13.8° | 29.1° | 0.11 eV |
| 1 | 12.3 bohr | 90° | 77.7° | 0.11 eV |

[1] Analogue structure of 1 with two chromophores.

The module of the EDTM obtained by TD-DFT for the monomer is 2.54 au. Modules of the two transitions in a truncated system from **1** with only two chromophores calculated by CSA are 1.80 and 3.15 au (see Figure S1 in the Supplementary Materials). These values agree very well with the ones predicted by cam-B3LYP, 1.71 and 3.09, respectively. For the total system **1** presenting three chromophores and $D_3$ symmetry, considering the CSA, three electronic transitions are allowed where two of them are degenerated and the EDTMs for transitions are given by the expression:

$$\vec{\mu}^{tA2} = \frac{\vec{\mu}_1^t + \vec{\mu}_2^t + \vec{\mu}_3^t}{\sqrt{3}} = \sqrt{3}\left|\vec{\mu}_1^t\right|\cos(\theta)\,\vec{k}, \tag{5}$$

$$\vec{\mu}^{tE1} = \frac{3\left|\vec{\mu}_1^t\right|\sin(\theta)}{2}\,\vec{i}, \tag{6}$$

$$\vec{\mu}^{tE2} = \sqrt{\frac{3}{2}}\left|\vec{\mu}_1^t\right|\sin(\theta)\,\vec{j}. \tag{7}$$

Figure 4 compares the EDTMs of A2 and E calculated by the CSA as well as EDTMs of A2, E and monomer calculated by TD-DFT.

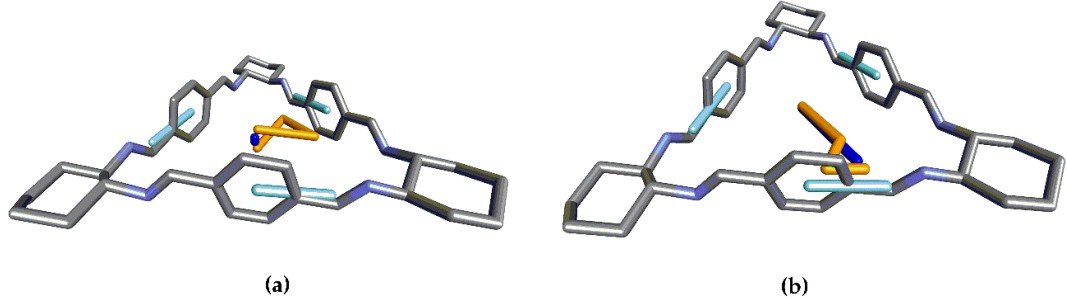

|  (a)  |  (b)  |

**Figure 4.** Representation of EDTMs for each chromophore (light blue, located in the chromophores) in structure **1** and total EDTMs (dark blue, located in the center of the system) for (**a**) A2 and (**b**) E transitions originated from the exciton coupling of main excitation of independent chromophores (gold, located in the center of the system).

Taking into account the structural measures collected in Table 1 and the expressions for the splitting and rotatory strengths, the results obtained for the different structures are shown in Table 2. When $\theta$ is close to 90°, discrepancies may arise between the relative intensity of the calculated transitions by CSA and TD-DFT. This issue will be further discussed below.

**Table 2.** Parameters obtained from CSA analysis of **1**. Rotatory strength values are given in ascending order of energy.

| Structure | Splitting CSA/TD-DFT | Module of EDTM CSA/TD-DFT | Rotatory Strength CSA/TD-DFT |
|---|---|---|---|
| **1** [1] | -/4.8026 eV | -/2.54 | -/- |
| **1** [2] | 0.22/0.20 | [3] 1.8:3.1/1.7:3.1 | [3] −334:334/−272:448 |
| **1** | 0.33/0.31 | 0.6:2.0/0.2:3.4 | [4] −1190/595/−363:332 |

[1] Analogue structure of **1** with only one chromophore, the energy displayed here corresponds with the most significant low energy electronic transition; [2] Analogue structure of 1 with two chromophores; [3] A:A. [4] A2:E.

### 3.2. Structure 2

The structural measures collected for CSA analysis of structure 2 are presented in Table 3, while two possible transitions of the independent chromophores, 2.1 and 2.2 are shown for structure 2 in Figure 5.

**Table 3.** Structural measures for CSA analysis of **2**.

| Structure | $R_{12}$ | $\omega$ | $\Theta$ | $V_{12}$/eV |
|---|---|---|---|---|
| 2.1 | 12.4 bohr | 90° | 99.7° | 0.05 eV |
| 2.2 | 12.4 bohr | 90° | 74.4° | 0.05 eV |

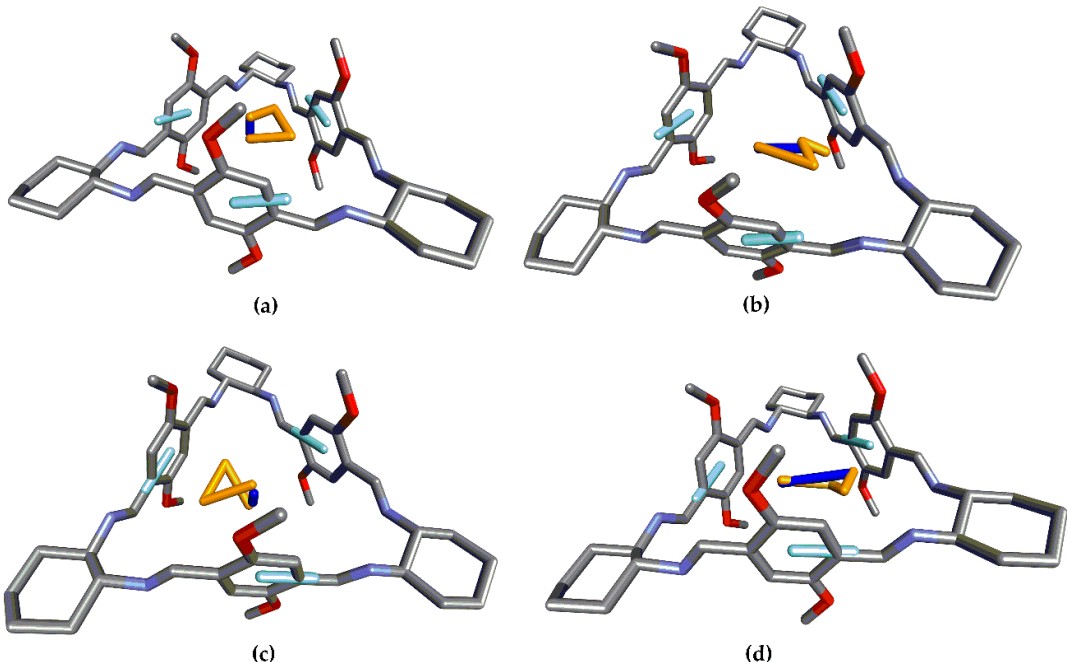

(a)      (b)

(c)      (d)

**Figure 5.** Representation of EDTMs for chromophores (light blue, located in the chromophores) in **2** and total EDTM (dark blue, located in the center of the system) for A2 transitions from 2.1 (**a**) and 2.2 (**c**) and E transitions from 2.1 (**b**) and 2.2 (**d**) originated form the exciton coupling of two main excitations of independent chromophores (gold, located in the center of the system).

Based on the measures collected in Table 3 and the expressions for the splitting and rotatory strengths, the results of the different structures are shown in Table 4.

**Table 4.** Parameters employed for CSA analysis of structure **2**. Rotatory strength values are given in ascending order of energy.

| Structure | Splitting CSA/TD-DFT | Module of EDTM CSA/TD-DFT | Rotatory Strength CSA/TD-DFT |
|---|---|---|---|
| 2.1 [1] | -/3.8879 eV | -/1.75 | -/- |
| 2.1 | 0.15/0.15 | [3] 0.6:1.7/0.6:2.26 | [3] 368:−184/649:−349 |
| 2.2 [2] | -/4.8181 eV | -/1.91 | -/- |
| 2.2 | 0.15/0.15 | [3] 0.8:1.6/0.58:2.57 | [3] −847:423/−665:325 |

[1] Analogue structure 2.1 with only one chromophore, the energy displayed here corresponds with the most significant low energy electronic transition; [2] Analogue structure 2.2 with only one chromophore, the energy displayed here corresponds with the most significant second low energy electronic transition; [3] A2:E. 2.1 refers to the exciton coupling originated from the lowest electronic transition of the monomer and 2.2 to the second lowest.

### 3.3. Structure 3

The structural parameters for CSA analysis of structure 3 are depicted in Table 5. On the other hand, Figure 6 shows structure 3 for two of the three possible transitions of the independent chromophores, 3.1 and 3.3. Transition 3.2 presents a $\theta$ c.a. 90°; consequently, the CSA would predict a negligible contribution to the ECD response from the EC of such transition.

**Table 5.** Structural measures for CSA analysis of 3.

| Structure | $R_{12}$ | $\omega$ | $\Theta$ | $V_{12}$/eV |
|---|---|---|---|---|
| 3.1 | 12.4 bohr | 90° | 77° | 0.02 eV |
| 3.2 | 12.4 bohr | 90° | c.a. 90° | 0.04 eV |
| 3.3 | 12.4 bohr | 90° | 94° | 0.03 eV |

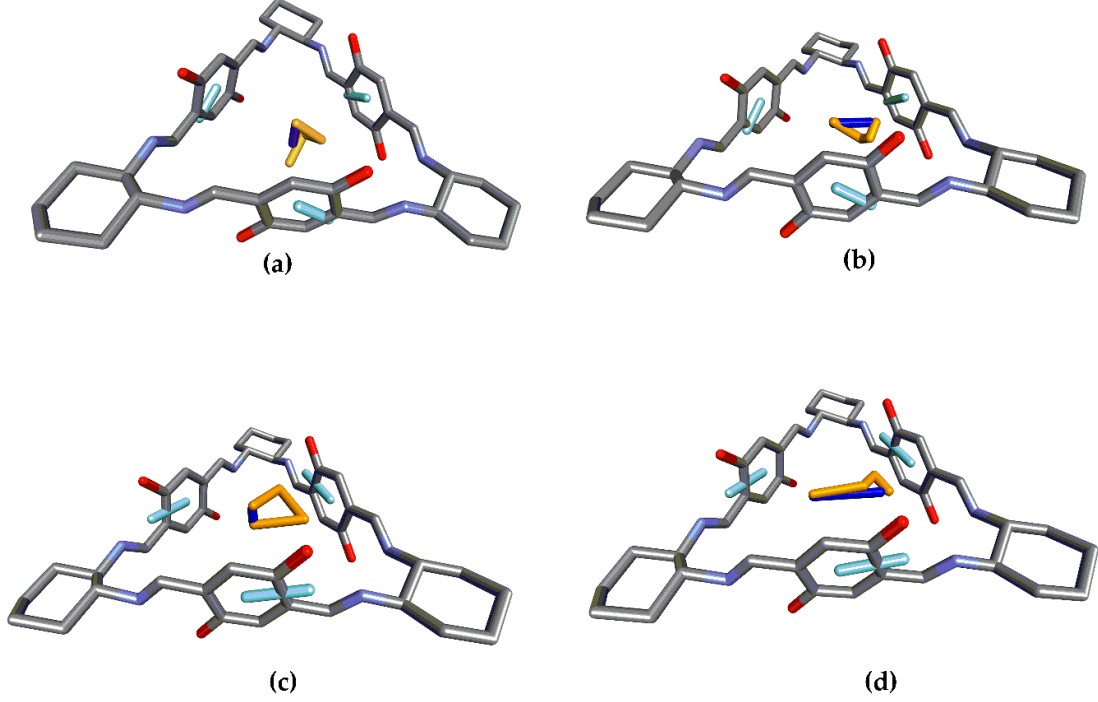

(a)  (b)

(c)  (d)

**Figure 6.** Representation of EDTMs for chromophores (light blue, located in the chromophores) in 3 and total EDTM (dark blue, located in the center of the system) for A2 transitions from (**a**) 3.1 and 3.3 (**c**) and transitions E from 3.1 (**b**) and 3.3 (**d**) originated from the exciton coupling of two of the three main excitations of the independent chromophores (gold, located in the center of the system).

Taking into account the values collected in Table 5 and the expressions for the splitting and rotatory strengths, the results obtained for the different structures are shown in Table 6.

**Table 6.** Parameters employed for CSA analysis of structure **3**. Rotatory strength values are given in ascending order of energy.

| Structure | Splitting CSA/TD-DFT | Module of EDTM CSA/TD-DFT | Rotatory Strength CSA/TD-DFT |
|---|---|---|---|
| 3.1 [1] | -/3.4007 eV | -/1.41 | -/- |
| 3.1 | 0.06/0.06 | [3] 1.3:1.0/1.09:1.50 | [3] −276:138/−801:304 |
| 3.3 [2] | -/5.7592 eV | -/2.05 | -/- |
| 3.3 | 0.09/0.14 | [3] 0.6:1.7/0.6:2.69 | [3] 313:−157/813:−535 |

[1] Analogue structure 3.1 with only one chromophore, the energy displayed here corresponds with the most significant low energy electronic transition; [2] Analogue structure 3.3 with only one chromophore, the energy displayed here corresponds with the most significant third low energy electronic transition; [3] A2:E. 3.1 refers to the exciton coupling originated from the lowest electronic transition of the monomer and 3.2 to the second lowest.

## 4. Discussion

The application of the CSA analysis for the truncated system 1 with two chromophores helps to visualize how the two independent chromophores interact as parallel or antiparallel to split into two electronic transitions with a defined orientation of the EDTM (Figure S1). The comparison of CSA and TD-DFT shows the reliability of the exciton coupling for the prediction of the main features of the ECD. In addition, CSA of 1 assist in understanding how the EDTM of the allowed transitions are oriented with respect to the system. This is of special interest when looking for chiroptical applications, for instance on the development of ordered monolayers for the construction of chiroptical surfaces [11]. In this study, for instance, we can observe from the CSA that the A2 transition oriented parallel to the $C_3$-axis has no comparable partner in the truncated system with only two chromophores. Notably, the EDTM and RS predicted with CSA are sufficient to reproduce the TD-DFT calculations well.

The chromophores in structure 2 present two significant electronic transitions with very distinct energies, 3.8879 eV and 4.8181 eV and relative orientations $\theta = 99.7$ and $\theta = 77.4$. Thus, CSA of the two transitions independently allows the prediction of the main features in two regions of the spectra. The CSA represented in Figure 5 shows how the module of the EDTMs associated with each transition can be tuned by the relative orientation of the electronic transition of the independent chromophore with respect to the molecular system $\theta$. Table 4 demonstrates how well the CSA compares with the TD-DFT predicted main parameters for the ECD.

Macrocycle 3 presents three significant electronic transitions with very distinct energies; however, the electronic transition 3.2 has a $\theta$ c.a. 90°. As the sign of the RS changes by exceeding this angle and CSA is only an approximation, in this case, prediction by the CSA is not recommended. Taking this into account, the prediction was performed considering the other two electronic transitions 3.1 and 3.3. Finally, the main features of the ECD could be obtained by the CSA as depicted in Table 6.

Remarkably, the $\theta$ values for the two transitions under consideration for both **2** and **3** are one higher and the other lower than 90°. Since this fact renders a flip in the sign of the RS, the main features of the predicted ECD spectra present a positive and a negative exciton couplet at the same time in the same spectra at different energies (Figure 7).

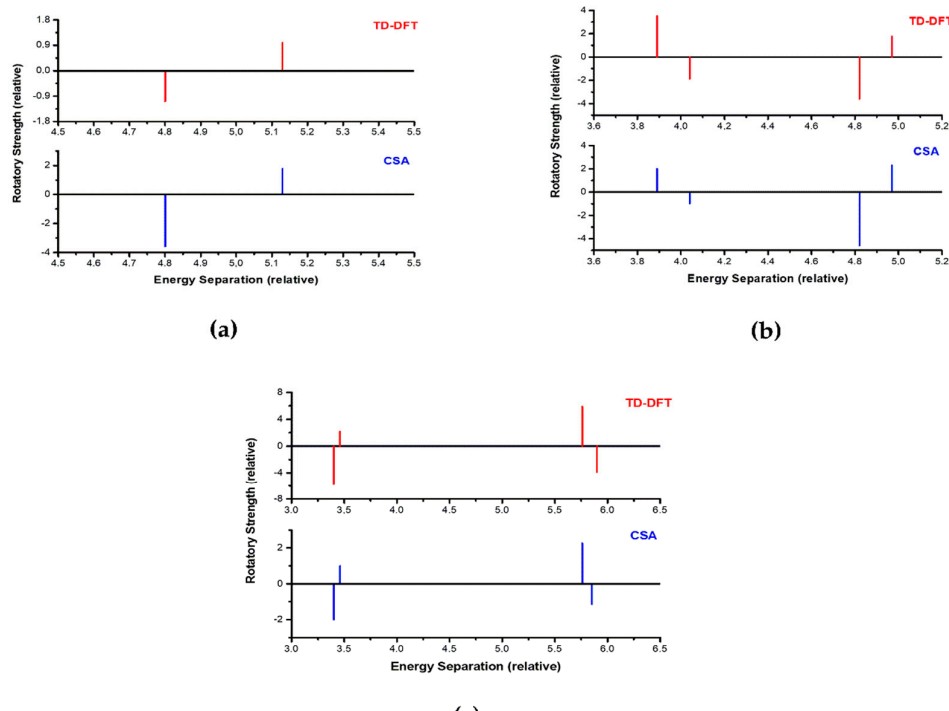

**Figure 7.** Comparison of TD-DFT (red) and CSA (blue) results for trianglimine systems (**a**) **1**; (**b**) **2;** and (**c**) **3**. Rotatory strength values are represented in relative intensities' energy as well as relative intensity of rotatory strength values resemble qualitatively those reported by Szymkowiak et al. [21].

## 5. Conclusions

The chiroptical symmetry analysis of trianglimines 1, 2, and 3 presented here shows how this methodology can be very useful for the design of chiroptical systems. In addition to predicting the main ECD signatures, CSA also assists with uncovering the mechanisms underlying such responses. On the other hand, ECD signatures obtained from the electronic transitions of independent chromophores were oriented by c.a. 90° respect to the main symmetry axis may not be predicted via CSA due to the flip of the sign of the RS about this angle and the approximation character of the model. We hope that this case study will help researchers to better understand the chiroptical responses for systems with high symmetry and to design systems for everyday chiroptical applications.

**Supplementary Materials:** The following are available online at http://www.mdpi.com/2073-8994/11/10/1245/s1, Figure S1 and coordinates for structures 1, 2 and 3.

**Author Contributions:** Conceptualization, J.L.A.-G. and Á.P.-G.; methodology, R.A.M.; validation, J.L.A.-G. and Á.P.-G; investigation, A.O. and R.P.-C.; writing—original draft preparation, A.O.; supervision, J.L.A.-G. and Á.P.-G.

**Funding:** This research was funded by Xunta de Galicia (ED431F 2016/005, GRC2019/24) and IBEROS (0245_IBEROS_1_E).

**Acknowledgments:** A.O. thanks Xunta de Galicia for a predoctoral fellowship. J.L.A.-G. thanks the Spanish Ministerio de Economía y Competividad for a "Ramón y Cajal" research contract. Authors are thankful to the Supercomputing Center of Galicia (CESGA) for generous allocation of computer time. The reviewers are acknowledged for their valuable suggestions to improve the manuscript.

**Conflicts of Interest:** The authors declare no conflict of interest.

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
