# Peer review of "Chiroptical Symmetry Analysis of Trianglimines: A Case Study"

_symmetry, doi:10.3390/sym11101245_

Round 1

Reviewer 1 Report

The present work by Ozcelik et al. applies a symmetry-adapted exciton chirality method for a few multichromophoric organic molecules presenting D3 symmetry. While a "visual" method to determine the chiral response in simplified systems would be interesting, the presentation of this work is highly confusing, even for experts in the field. Moreover, the Introduction does not put the problem in the context of existing literature; the discussion is very limited and does not consider the various limitations of the method; and the results are not compared with experiments (which are present in the literature!). Therefore, I cannot recommend publication of this paper in its present form.

Major issues:

The (very brief) Introduction is confusing, and fails to present the exciton chirality method and its extensions in the context of the existing literature. In particular, the cited literature is limited to a few examples and self-citations of the Authors, and it completely neglects all attempts in the literature at extending the exciton model to ab initio descriptions of the excitation. The Authors state that for ab initio methods "when the system of interest is large, the necessary computational and time resources may be a limitation".

However, (i) their method is necessarily based on ab initio estimations of the transition dipole moments, and (ii) they ignore the efforts by several groups in developing an exciton model based on ab initio calculations, which greatly reduce the computational cost. The Authors cite experimental results for the systems studied in this work, however they do not compare their results with the experiments.

This makes readers think that the method is not accurate enough to compare with experiments. The Authors estimate the exciton couplings with a very crude approximation, without explaining that it is an approximation. In addition, eqs. 3/4 are only valid neglecting the intrinsic magnetic transition dipole moments of the monomers. The presentation of results and the discussion are very confused, and do not allow the reader to understand what is the utility of the CSA with respect to building the exciton Hamiltonian matrix (as in the matrix method), and solving for the energies and rotatory strengths of the exciton states. What is missing is a clear "visual" rule that allows one to predict the sign of a couplet as is easily done with the 2-chromophore ECM. The results for system 1 are quite different between the CSA and TDDFT. In particular, the rotatory strength of the A2 transition is not double the rotatory strength of the E transition. This suggests that either there is intrinsic chirality in the monomer transitions (i.e. an intrinsic magnetic transition dipole) or there are higher lying transitions that couple with the transition of interest. Either way, this point should be commented. The tables and figures are not clear. For example, Table 2 is difficult to understand, and the caption does not explain the Table (It is copied from Table 1). This indicates a complete lack of care in presenting results and making them understandable to the readers. The English is quite poor, with several grammar and spelling errors, such as "undistinguishable", "in a qualitatively way", "helps to visualized", "for symmetry D3 is necessary 90" ; some abbreviations such as RS (rotatory strength?) are not defined.

Author Response

Reviewer 1

The present work by Ozcelik et al. applies a symmetry-adapted exciton chirality method for a few multichromophoric organic molecules presenting D3 symmetry. While a "visual" method to determine the chiral response in simplified systems would be interesting, the presentation of this work is highly confusing, even for experts in the field. Moreover, the Introduction does not put the problem in the context of existing literature; the discussion is very limited and does not consider the various limitations of the method; and the results are not compared with experiments (which are present in the literature!). Therefore, I cannot recommend publication of this paper in its present form.

Major issues:

The (very brief) Introduction is confusing, and fails to present the exciton chirality method and its extensions in the context of the existing literature. In particular, the cited literature is limited to a few examples and self-citations of the Authors, and it completely neglects all attempts in the literature at extending the exciton model to ab initio descriptions of the excitation.

We have revised the introduction including several references. We hope that now it is clearer to the reader. Particularly, the following sentences have been added: “An object is chiral when it is non-superimposable with its mirror-image.[1] Examples presenting this property vary from macroscopic such as gloves or snails to microscopic e. nanoparticles[2] and molecules[3] among others. Furthermore, two chiral mirror-image molecules are called enantiomers. While the interaction between two enantiomeric systems and any achiral medium is indistinguishable, the behavior may differ when the two enantiomers interact with a chiral medium; giving rise to chiroptical responses when interacting with chiral light.[4]” “Ab initio simulations have been shown to have great relevance for the prediction of chiroptical responses.[13] However, even simplified TD-DFT techniques have been developed for large systems,[14,15] the necessary computational and time resources may be a limitation. As a qualitative alternative, the application of the exciton chirality method[16] has been considered to be practical for systems incorporating two chromophores for instance molecular tweezers used as chiroptical proof of natural amines.[17] Nevertheless, the presence of more chromophores may be desired for designing powerful chiroptical systems.[18] In this respect, we previously revised the chiroptical symmetry analysis (CSA) for systems presenting high symmetry.[19] The main scope of this approach is to serve as a useful tool for developing particular systems rather than predicting experimental spectrum of a system. On the other hand, different parameters (i.e. rotatory strength or energy difference between allowed transitions) can be calculated by the CSA. Considering the conformational stability typically presented by trianglimines as well as taking advantage of the experimental and simulated ECD reported by Szymkowiak et al,[20] in this work we perform the chiroptical symmetry analysis for 1, 2, and 3 (Figure 1) as a case and intent to draw some conclusions about the reliability of this strategy on the design of tailored chiroptical systems.”

The Authors state that for ab initio methods "when the system of interest is large, the necessary computational and time resources may be a limitation". However, (i) their method is necessarily based on ab initio estimations of the transition dipole moments, and (ii) they ignore the efforts by several groups in developing an exciton model based on ab initio calculations, which greatly reduce the computational cost.

We agree with the reviewer on that the CSA requires TD-DFT calculations. However, the only necessary calculation is for the independent chromophores. This reduces significantly the size of the system to compute, and consequently the time of calculation. To clarify this point, we have implemented the following text “It is noteworthy that this approximation is only valid when the electronic transition of the independent chromophore presents negligible magnetic transition dipole moment (MTDM, for a more thorough explanation see reference [19]). Other studies have developed ab initio based exciton chirality methods.[24]”

The Authors cite experimental results for the systems studied in this work, however they do not compare their results with the experiments. This makes readers think that the method is not accurate enough to compare with experiments.

Indeed there is no an explicit figure comparing the experimental ECD with the sipulated by CSA. To avoid misunderstandings we have included the following sentence “Considering the conformational stability typically presented by trianglimines as well as taking advantage of the experimental and simulated ECD reported by Szymkowiak et al,[20] in this work we perform the chiroptical symmetry analysis for 1, 2, and 3 (Figure 1) as a case study and intent to draw some conclusions about the reliability of this strategy on the design of tailored chiroptical systems.” and “Particularly, the cam-B3LYP functional was chosen since the simulated spectra showed good agreement with the reported experimental ECD spectra.[20]”. Additionally, to visualize the agreement between CSA and TD-DFT, Figure 7 has been added to the manuscript.

The Authors estimate the exciton couplings with a very crude approximation, without explaining that it is an approximation. In addition, eqs. 3/4 are only valid neglecting the intrinsic magnetic transition dipole moments of the monomers.

The following paragraph has been added to the manuscript: “It is noteworthy that this approximation is only valid when the electronic transition of the independent chromophore presents negligible magnetic transition dipole moment (MTDM, for a more thorough explanation see reference [19]). Other studies have developed ab initio based exciton chirality methods.[24]”

The presentation of results and the discussion are very confused, and do not allow the reader to understand what is the utility of the CSA with respect to building the exciton Hamiltonian matrix (as in the matrix method), and solving for the energies and rotatory strengths of the exciton states. What is missing is a clear "visual" rule that allows one to predict the sign of a couplet as is easily done with the 2-chromophore ECM.

The modified manuscript involves “Materials and Methods” and “Results” sections. We hope that this facilitates the reader to follow our explanations. In particular, the following text has been introduced to make more visual the CSA concerning the relative orientation of the interacting chromophores: “Remarkably, the θ values for the two transitions under consideration for both 2 and 3 are one higher and other lower than 90°. Since this fact renders a flip in the sign of the RS, the main features of the predicted ECD spectra present a positive and a negative exciton couplets at the same time in the same spectra at different energies." Additionally, Figure 2 has been modified to incorporate w.

The results for system 1 are quite different between the CSA and TDDFT. In particular, the rotatory strength of the A2 transition is not double the rotatory strength of the E transition. This suggests that either there is intrinsic chirality in the monomer transitions (i.e. an intrinsic magnetic transition dipole) or there are higher lying transitions that couple with the transition of interest. Either way, this point should be commented.

We agree with the reviewer, to clarify this point we have included in the text the following sentence “Discrepancies between the relative intensity of the calculates transitions by CSA and TD-DFT may arise when θ is close to 90 as it will be discussed below.”

The tables and figures are not clear. For example, Table 2 is difficult to understand, and the caption does not explain the Table (It is copied from Table 1). This indicates a complete lack of care in presenting results and making them understandable to the readers.

The authors apologize for the misleading information. The figures and tables have been modified accordingly.

The English is quite poor, with several grammar and spelling errors, such as "undistinguishable", "in a qualitatively way", "helps to visualized", "for symmetry D3 is necessary 90" ; some abbreviations such as RS (rotatory strength?) are not defined.

The manuscript has been revised and all corrections have been highlighted in the final version of the manuscript.

Reviewer 2 Report

Edits

L42 relevant -> relevance

General comments

This paper applies the principles of CSA applied to a system of D3 symmetry, the formalism for which was laid out in a previous paper by many of the same authors (see ref 14). The paper is well written. This works seeks to analyze some trianglimines to make comparisons between the exciton chirality (EC) method and TD-DFT calculations, with the goal of understanding the ECD and relate that to larger systems. The problem with applying EC analysis to large systems, is that it is typically best to do quantum calculations anyway, because a large structure will have a great deal of conformational freedom. Also there are techniques to calculate the ECD of very large molecules. These should be mentioned (see J. Chem. Phys. 126, 134102 (2007) and Christoph Bannwarth, Stefan Grimme, Computational and Theoretical Chemistry, 1040–1041,(2014)Pages 45-53)

What is also important with calculating ECD is the overall character of the resulting spectrum that is produced. If transitional energies or rotational strengths are incorrect, then you can get cancelation in the wrong place, and change the overall appearance of the spectrum. Also the calculated ECD does match the experimental ECD for similar molecules (as shown in ref 16) but the calculations are not perfect. How close do the EC predictions come? Can you simulate them and compare?

Author Response

Edits

L42 relevant -> relevance

It has been corrected.

General comments

This paper applies the principles of CSA applied to a system of D3 symmetry, the formalism for which was laid out in a previous paper by many of the same authors (see ref 14). The paper is well written. This works seeks to analyze some trianglimines to make comparisons between the exciton chirality (EC) method and TD-DFT calculations, with the goal of understanding the ECD and relate that to larger systems.

The problem with applying EC analysis to large systems, is that it is typically best to do quantum calculations anyway, because a large structure will have a great deal of conformational freedom. Also there are techniques to calculate the ECD of very large molecules. These should be mentioned (see J. Chem. Phys. 126, 134102 (2007) and Christoph Bannwarth, Stefan Grimme, Computational and Theoretical Chemistry, 1040–1041,(2014)Pages 45-53)

In order to address these aspects, the abstract now reads: “In this regard, the exciton chirality method has been often used in conformationally stable systems incorporating at least two independent chromophores.” and the introduction now reads: “Ab initio simulations have been shown to have great relevance for the prediction of chiroptical responses.[13] However, even simplified TD-DFT techniques have been developed for large systems,[14,15] the necessary computational and time resources may be a limitation.”

What is also important with calculating ECD is the overall character of the resulting spectrum that is produced. If transitional energies or rotational strengths are incorrect, then you can get cancelation in the wrong place, and change the overall appearance of the spectrum. Also the calculated ECD does match the experimental ECD for similar molecules (as shown in ref 16) but the calculations are not perfect. How close do the EC predictions come? Can you simulate them and compare?

To clarify this point, the following text has been included “Particularly, the cam-B3LYP functional was chosen since the simulated spectra showed good agreement with the reported experimental ECD spectra.[20]” Additionally, Figure 7 has been included in the manuscript.

Reviewer 3 Report

In this manuscript, the authors describe the use of so-called chiroptical symmetry analysis for prediction of exciton coupling in ECD spectra of some trianglimines. This work is closely related to the previous contribution from the same group, published in Molecules. In this manuscript , the authors focused on the practical applications of the method to study the real molecular systems.

Overall the data is of good quality; the central question of the manuscript is of relevance to everyone working with chiral substances. The conclusions are justified. Although the exciton chirality method is known for decades, the methodology proposed by the authors can help in rational planning of new molecular and especially supramolecular systems characterized by high chiroptical response.

Overall, this is an interesting paper that should be published in Symmetry after considering some minor points.

1) In Introduction the authors focused mostly on the phenomenon. I believe that a few sentences about objects (along with literature references) would be advisable.

2) The authors should avoid too categorical statements such as " extremely sensitive". While in some cases this is true, in general such a statement is too far-reaching.

3) The authors compare only theoretical results. Any reference to the experiment would be advisable. Similarly, approximating the obtained rotatory strengths with a Gaussian function would give a better idea of the suitability of a given method than just the numbers.

4) The table captions look weird in my eyes.

Author Response

In this manuscript, the authors describe the use of so-called chiroptical symmetry analysis for prediction of exciton coupling in ECD spectra of some trianglimines. This work is closely related to the previous contribution from the same group, published in Molecules. In this manuscript, the authors focused on the practical applications of the method to study the real molecular systems.

Overall the data is of good quality; the central question of the manuscript is of relevance to everyone working with chiral substances. The conclusions are justified. Although the exciton chirality method is known for decades, the methodology proposed by the authors can help in rational planning of new molecular and especially supramolecular systems characterized by high chiroptical response.

Overall, this is an interesting paper that should be published in Symmetry after considering some minor points.

1) In Introduction the authors focused mostly on the phenomenon. I believe that a few sentences about objects (along with literature references) would be advisable.

We have added to the introduction a more general definition with related examples and references. The text now reads: “An object is chiral when it is non-superimposable with its mirror-image.[1] Examples presenting the this property vary from macroscopic such as gloves or snails to microscopic i.e. nanoparticles[2] and molecules[3] among others. Furthermore, two chiral mirror-image molecules are called enantiomers. While the interaction between two enantiomeric systems and any achiral medium is indistinguishable, the behavior may differ when the two enantiomers interact with a chiral medium; giving rise to chiroptical responses when interacting with chiral light.[4]”

2) The authors should avoid too categorical statements such as " extremely sensitive". While in some cases this is true, in general such a statement is too far-reaching.

The statements have been removed.

3) The authors compare only theoretical results. Any reference to the experiment would be advisable. Similarly, approximating the obtained rotatory strengths with a Gaussian function would give a better idea of the suitability of a given method than just the numbers.

To clarify this aspect, the following text has been added to the manuscript “Considering the conformational stability typically presented by trianglimines as well as taking advantage of the experimental and simulated ECD reported by Szymkowiak et al,[20] in this work we perform the chiroptical symmetry analysis for 1, 2, and 3 (Figure 1) as a case and intent to draw some conclusions about the reliability of this strategy on the design of tailored chiroptical systems.” and “Particularly, the cam-B3LYP functional was chosen since the simulated spectra showed good agreement with the reported experimental ECD spectra.[20]”. Additionally, Figure 7 comparing the RS calculated by CSA and TD-DFT has been included in the manuscript. Since we are compared the RS we find more useful to use bars instead of Gaussian curves.

4) The table captions look weird in my eyes.

The caption of each table has been corrected in order to clarify the results.

Round 2

Reviewer 1 Report

The Authors have somewhat addressed my previous point. However, there are still several issues with the current manuscript.

The Introduction could be improved by explaining how the exciton chirality method works, and how the method developed by the Authors extends the ECM. The Methods section could also be improved by drawing analogies with the ECM and giving a qualitative rule for predicting the sign of a CD couplet. How does the CD couplet predicted by CSA compare to the couplet predicted on two chromophores by the ECM? Moreover, the equations are somewhat confusing. Eqs. (3) and (4) are the same except a -1/2 factor, yet they look different. It is advisable to use only eq. (3), because the rotatory strength of the other states can be derived from the one of the A state. I still do not understand why the Authors do not draw a comparison with the experiments. This could help highlighting strengths and weaknesses of the method. In Table 1, I assume that the Authors refer to the analogue of structure 1 with TWO chromophores. Also, I assume that in Tables 2/4/6 the rotatory strengths are given in ascending order of energy; this could be clarified. The English is still poor, and some grammar error make the comprehension difficult. e.g. " even simplified TD-DFT [...]" -> "even THOUGH simplified TD-DFT [...]", "the rotatory strengths associate to each allowed transition", " ECD signatures originated from "

Author Response

Major issues:

The Authors have somewhat addressed my previous point. However, there are still several issues with the current manuscript.

We are happy to know that the referee found satisfactory the revised version of our manuscript after attending her/his suggestions.

The Introduction could be improved by explaining how the exciton chirality method works, and how the method developed by the Authors extends the ECM.

The Introduction now reads as follows to address the abovementioned aspects: “As a qualitative alternative, the application of the exciton chirality method (ECM)[16] has been considered to be practical for systems incorporating two identical chromophores, for instance molecular tweezers used as chiroptical proof of natural amines.[17]. When two identical chromophores are in close vicinity, in phase and out of phase excitations arise. If the mutual orientation between the chromophores is chiral, the rotatory strength (RS) of the two transitions may be of opposite sign. The sign of the RS associated with the less energetic transition determines the sense of chirality.[18]”

The Methods section could also be improved by drawing analogies with the ECM and giving a qualitative rule for predicting the sign of a CD couplet.

To clarify this point, the following text has been added to the manuscript “Like EC, CSA also provides the sense of chirality of the responding system.”

How does the CD couplet predicted by CSA compare to the couplet predicted on two chromophores by the ECM?

In order to clarify this point, the following statement has been added: “It should be noted that the application of CSA for systems bearing two chromophores is equal to that of ECM.”

Moreover, the equations are somewhat confusing. Eqs. (3) and (4) are the same except a -1/2 factor, yet they look different. It is advisable to use only eq. (3), because the rotatory strength of the other states can be derived from the one of the A state.

The equation 4 now reads as: “”

I still do not understand why the Authors do not draw a comparison with the experiments. This could help highlighting strengths and weaknesses of the method.

The authors of the present manuscript did exclusively a theoretical study to compare with the experimental results from Szymkowiak et al. To better clarify this aspect, the text now reads: “Structures 1, 2, and 3 presenting three chromophores were hereby studied by CSA. Szymkowiak et al. demonstrated that TD-DFT calculations resemble well the experimental ECD spectra of those compounds.[21] Therefore, as mentioned above, DFT simulations were performed and compared with those obtained by CSA.” and “Figure 7. Comparison of TD-DFT (red) and CSA (blue) results for trianglimine systems (a) 1, (b) 2 and (c) 3. Rotatory strength values are represented in relative intensities. Energy as well as relative intensity of rotatory strength values resemble qualitatively those reported by Szymkowiak et al.[21].

In Table 1, I assume that the Authors refer to the analogue of structure 1 with TWO chromophores.

We apologize for the misleading caption. It has been modified accordingly.

Also, I assume that in Tables 2/4/6 the rotatory strengths are given in ascending order of energy; this could be clarified.

For the sake of clarity, the following statement has been included to each table: “Rotatory strength values are given in ascending order of energy.”

The English is still poor, and some grammar error make the comprehension difficult. e.g. " even simplified TD-DFT [...]" -> "even THOUGH simplified TD-DFT [...]", "the rotatory strengths associate to each allowed transition", " ECD signatures originated from "

The manuscript has been revised and all modifications have been highlighted.
